# Understanding the underlying drivers of obesity in Africa: a scoping review protocol

Anelisa Jaca [1], Chinwe Iwu [2], Solange Durão,[1] Adelheid W Onyango,[3] Charles Shey Wiysonge [1,4,5]

[1]Cochrane South Africa, South African Medical Research Council, Tygerberg, South Africa
[2]Department of Nursing and Midwifery, Faculty of Medicine and Health Sciences, Stellenbosch University, Cape Town, South Africa
[3]Department of Nutrition for Health and Development, World Health Organization, Brazzaville, Congo
[4]School of Public Health and Family Medicine, University of Cape Town, Cape Town, South Africa
[5]Department of Global Health, Stellenbosch University, Cape Town, South Africa

**Correspondence to**
Dr Anelisa Jaca;
anelisaj@gmail.com

## ABSTRACT

**Introduction** The worldwide prevalence of obesity and overweight has doubled since 1980, such that approximately a third of the world's population is reported as obese or overweight. Obesity rates have increased in all ages and both sexes irrespective of geographical area, ethnicity or socioeconomic status. Due to the high prevalence, related health consequences and costs of childhood and adult obesity, there is a need to comprehensively identify and assess the major underlying drivers of obesity and overweight in the African context.

**Methods and analysis** This scoping review will be carried out as per the methodological outline by Arksey and O'Malley. The search strategy will be developed and search performed in the Scopus and PubMed electronic databases. In the first search, we will identify concepts that are used as an equivalent to obesity and overweight. Subsequently, we will search for studies comprising of search terms on the underlying factors that drive the development of obesity and overweight. Lastly, we will check reference lists for additional publications. Abstracts and full-text studies will independently be screened by two authors.

**Ethics and dissemination** The proposed study will generate evidence from published data and hence does not require ethics approval. Evidence generated from this review will be disseminated through journal publications and conference presentations.

## INTRODUCTION

Obesity is an intricate, multifactorial and preventable health issue currently affecting, together with overweight, over a third of the world's population.[1] The prevalence of obesity is seen across all ages, populations, ethnic groups and socioeconomic status.[2] Obesity was once considered a high-income country health problem but it is now also commonly seen in low-income and middle-income countries.[3 4] In low-income countries, this condition is more predominant among middle-aged adults, especially women of childbearing age (15 and 49 years), while it affects individuals of all ages and genders in high-income countries.[5] In 2016, over 1.9 billion adults were overweight and of those individuals, more than 650 million were obese, worldwide.[6] Approximately 40 million

children under the age of 5 years and 340 million adolescents were reported to be obese between 2016 and 2018, globally.[7] The number of children, under the age of 5 years, suffering from obesity or overweight in Africa was estimated at 9.5 million in the year 2018.[8] It is estimated that 20% of the world's adult population will be obese and 38% overweight by 2030 if this health issue is not dealt with.[1]

The WHO defines obesity as a surplus of fat that can affect one's health. The commonly used measure of obesity is the body mass index (BMI), calculated by dividing the body weight in kilograms by the square of height in metres.[9] The WHO defines a normal BMI range as 18.5 to 24.9 suggesting that individuals with BMI greater than 25 kg/m$^2$ and 30 kg/m$^2$ are considered overweight and obese, respectively.[10] Obesity affects nearly all physiological functions of the body and hence increases the risk of developing conditions such as cardiovascular (eg, heart failure) and metabolic (type 2 diabetes mellitus, hypertension and stroke) diseases.[10–12]

Obesity is predominantly associated with the overconsumption of foods with high sugar, calories and fats through the overconsumption of processed, inexpensive and poor quality foods that are not nutrient dense.[13] The excess food that is consumed is changed into triglycerides which are stored in adipose tissues, hence increasing body fat and ultimately resulting in weight gain.[14 15] Additionally, the nutrition

transition, which encompasses changes from predominantly plant-based diets to those predominantly meat-based, is highly associated with weight gain and illnesses related to it.[16 17] Therefore, healthy eating behaviours can play a major role in reducing the prevalence of obesity and its related diseases.[18]

### Strengths and limitations of this study

► This scoping review will synthesise evidence on all factors related to the drivers of obesity in Africa.
► This study will employ a rigorous methodology which involves an established research framework, a search strategy and a selection process.
► Our review will potentially not include all research papers related to this theme area as some may not be accessible or published.

Alongside the rise in consumption of processed foods there has also been an increase in sedentary behaviours in Africa.[19 20] This may have contributed to increased rates of obesity as weight gain is associated with a positive energy balance, that is, calorie intake is greater than expenditure.[21] Literature has shown that obesity rates in adults are generally significantly higher in sedentary compared with active individuals.[22] A decrease in daily energy expenditure is associated with weight gain while high levels of physical activity significantly mitigate weight gain.[23 24] Physical activity is defined as any bodily movement that demands energy use, e.g., walking, cycling, sport activities and doing house chores. On the other hand, physical inactivity or sedentary behaviour is a lifestyle identified by an energy expenditure less than 1.5 metabolic equivalents while in a sitting, reclining or lying position.[25] In addition to dietary and sedentary patterns, other contributing factors to the high prevalence of obesity include smoking, alcohol consumption, urbanisation and commonly used medications, namely, psychotropic medications, diabetic treatments, antihypertensives, steroid hormones and contraceptives.[26]

## Study rationale

There is growing acknowledgement that it will be challenging to practice healthy lifestyles unless we understand the factors that drive and contribute to unhealthy behaviours. Factors including diet, level of physical activity and age have been widely reported to influence obesity. Obesity increases with age which may be explained by decreases in physical activity and metabolic activity in older adults. The more we understand how these, or other factors drive obesity in Africa, the better we will be able to address it. Without proper understanding of the drivers of obesity in this setting, many individuals will not only suffer from obesity-related conditions but from costs sustained during treatment. Since obesity not only contributes to poor health but also affects people's psychological and social well-being, addressing it requires understanding and appreciating our biology, behaviour, environment and culture. There is a need to better understand the various factors that motivate our eating habits and sedentarity, driving obesity. To date, no synthesised evidence exists that investigated the underlying drivers of obesity in Africa. We will therefore conduct a narrative synthesis of the underlying drivers of obesity in the African setting.

## Study objectives

The objective of this scoping review is to chart the existing evidence on the underlying drivers of obesity and overweight in the African context.

## Methodology

A scoping review is used to gather emerging evidence in various fields with relevance to time, population (eg, country or context), source (eg, peer-reviewed or grey literature) and origin (eg, healthcare discipline or academic field). This study design is suitable for addressing broad concepts, questions that seek to assess the effects of interventions to inform practice, policy and research. We will thus conduct a scoping review, for studies published by end of March 2020, in order to map the crucial concepts and bring together the existing evidence on the underlying drivers of obesity in the African context. We will carry out this scoping review employing a predefined protocol as per the Arksey and O'Malley methodological outline.[27] This outline includes identifying the research question, searching for relevant studies, selecting studies, charting and collating data, summarising and reporting results. This scoping review will be reported as per the Preferred Reporting Items for Systematic Reviews and Meta-Analyses extension for Scoping Reviews (PRISMA-ScR) checklist.[28]

## Defining the research question

This scoping review will identify the drivers of obesity in the African setting. The research question: 'What are the underlying drivers of obesity among children and adults in Africa' was framed using the Population, Exposure, Concept and Context element (table 1).

## Inclusion criteria

We will include studies of any design that focus on drivers or risk factors of obesity. These drivers may include dietary factors (eg, overconsumption of high-energy/low nutrient, processed and obesogenic foods), sedentary behaviours (eg, physical inactivity or increased sedentary behaviours), smoking status, alcohol consumption, urbanisation and commonly used medications (eg, psychotropic medications, diabetic treatments, antihypertensives, steroid hormones

**Table 1** PECC element for defining the eligibility criteria of the studies for the research question

| Population | Exposure | Concept | Context |
|---|---|---|---|
| Children (between ages 2 and 12 years), adolescents (13 to 17 years), young adults (18 years to 35 years), older adults (36 to 55 years) and the elderly (56 years and older) | Risk factors or drivers of obesity and overweight | Obesity (BMI greater than or equal 30 kg/m$^2$) Overweight (BMI greater than or equal to 25 kg/m$^2$) | Africa |

BMI, body mass index; PECC, Population, Exposure, Concept and Context.

**Table 2** Initial search strategy developed for PubMed for the scoping review on the underlying drivers of obesity and overweight among children and adults in Africa

| Search # | Search texts and syntaxes |
|---|---|
| #1 | obesity OR obese OR overweight OR body mass index OR fat OR fatness |
| #2 | physical inactivity OR physically inactive OR sedentar* |
| #3 | (((ALGERIA) OR (ANGOLA) OR (BENIN) OR (BOTSWANA) OR (BURKINA FASO) OR (BURUNDI) OR (CAMEROON) OR (CENTRAL AFRICAN REPUBLIC) OR (CHAD) OR (COMOROS) OR (CONGO) OR (DEMOCRATIC REPUBLIC CONGO) OR (DJIBOUTI) OR (EGYPT) OR ((EQUATORIAL GUINEA) OR 'EQUATORIAL GUINEA') OR (ERITREA) OR (ETHIOPIA) OR (GABON)) OR ((GAMBIA) OR (GHANA) OR (GUINEA) OR ((GUINEA BISSAU) OR 'GUINEA BISSAU') OR (IVORY COAST) OR ((COTE D'IVOIRE) OR 'COTE D'IVOIRE') OR ((COTE IVOIRE) OR 'COTE IVOIRE') OR (KENYA) OR (LESOTHO) OR (LIBERIA) OR (LIBYA) OR (LIBIA) OR (JAMAHIRIYA) OR (JAMAHIRYIA) OR (MADAGASCAR) OR (MALAWI) OR (MALI) OR (MAURITANIA) OR (MAURITIUS) OR (MOROCCO)) OR ((MOZAMBIQUE) OR (MOCAMBIQUE) OR (NAMIBIA) OR (NIGER) OR (NIGERIA) OR (REUNION) OR (RWANDA) OR ((SAO TOME) OR 'SAO TOME') OR (SENEGAL) OR (SEYCHELLES) OR ((SIERRA LEONE) OR 'SIERRA LEONE') OR (SOMALIA) OR ((SOUTH AFRICA) OR 'SOUTH AFRICA') OR ((ST HELENA) OR 'ST HELENA') OR (SUDAN) OR (SWAZILAND) OR ESWATINI OR (TANZANIA) OR (TANGANYIKA) OR (TOGO) OR (TUNISIA)) OR ((UGANDA) OR ((WESTERN SAHARA) OR 'WESTERN SAHARA') OR (ZAIRE) OR (ZAMBIA) OR (ZIMBABWE) OR (AFRICA(MH)) OR (SOUTH* AND AFRICA*) OR (WEST* AND AFRICA*) OR (EAST* AND AFRICA*) OR (NORTH* AND AFRICA*) OR (CENTRAL* AND AFRICA*) OR (SUB SAHARAN AFRICA*) OR (SUBSAHARAN AFRICA*) OR (AFRICA*))) NOT (((GUINEA PIG*) OR 'GUINEA PIG*') OR ((ASPERGILLUS NIGER) OR 'ASPERGILLUS NIGER')) |
| #4 | #1 AND #2 AND #3 |

and contraceptives). Furthermore, the studies must have assessed and reported outcomes such as the type of food consumed, type of medication taken, sedentary behavioural patterns and prevalence of obesity and overweight. This review will include obese and overweight individuals of all ages, that is, children (between ages 2 and 12 years), adolescents (13 to 17 years), young adults (18 to 35 years), older adults (36 to 55 years) and the elderly (56 years and older) and both genders (male and female). We will define overweight adults as those with a BMI greater than or equal to $25 \, kg/m^2$, while those with a BMI greater than or equal to $30 \, kg/m^2$ will be considered obese. In children between the ages of 5 and 19 years, overweight will be defined as a BMI >+1 SD equivalent to $25 \, kg/m^2$ while those with a BMI >+2 SD equivalent to $30 \, kg/m^2$ will be deemed obese.[29]

## Selection of studies relevant to the research question
### Search strategy
We will conduct a comprehensive literature search in Scopus and PubMed databases. No language or date restrictions will be used. In our search, we will combine the key terms such as 'obesity OR obese OR overweightness OR overweight OR weight gain' AND physical inactivity OR physically inactive OR sedentar* (table 2). The first 100 search results from each database will be reviewed by the researchers to assess validity of the search strategy. When agreement is reached about the initial search strategy, the first 200 abstracts will be screened by two authors (AJ and CI) on concepts potentially eligible for inclusion in the second search step. In the event of full agreement between the two authors on potentially eligible concepts, AJ will screen the rest of the abstracts. In the event of disagreement, CI will screen the abstracts until full agreement is reached. After all abstracts have been screened, AJ and CI will discuss all potentially eligible search terms and select words to be included in the subsequent search step. After having selected the different search terms, AJ and CI will develop a search query in PubMed and then apply it to other databases. Following the searches, the two authors will also search reference lists of relevant studies for other eligible articles.

## Screening and selection
Two review authors (AJ and CI) will independently screen the titles and abstracts of articles identified in the search output for potentially eligible studies. The authors will obtain the full-text articles for any study that is considered potentially eligible and then independently appraise the full text of each potentially eligible study and classify it as either included or excluded. We will provide reasons for excluding potentially eligible studies from the review. Disagreements among the two authors during the screening and study selection process will be resolved through discussion and arbitration by a third author.

## Data charting and extraction
Data will be extracted independently by two authors (AJ and CI) from the eligible studies and differences resolved by a third author (CSW). Data extracted will comprise study characteristics (year the study was conducted/published, country of origin, study design, number of study arms and comparators); participants (children and adults); setting (African countries) and drivers or risk factors of obesity. These will include demographic (age and gender), environmental (suburb and rural areas), food availability (high calories and energy dense food), socioeconomic (education level, employment status and income level), behavioural (eg, physical inactivity, diet and alcohol consumption) and medication-related drivers.

## Patient and public involvement
This study will employ publicly available data and hence will not involve any patients and the public.

## Collating, summarizsing and reporting results of the review
A PRISMA flow diagram will be used to report the final number of included studies in the review. We will synthesise study findings using narrative descriptions based on subjects that emerge from the extracted data. The results will be compared and consolidated through consensus between two authors (AJ and CI), where both quantitative and qualitative aspects of research evidence will be

addressed. We will discuss and present data obtained from obesity-related outcomes reported in each study.

## Expert consultation

In order to confirm our findings and interpretations, an expert in the field of nutrition and obesity will be approached for consultation.

## Ethics and dissemination

The methodology applied to carry out this review will comprise reviewing and collecting evidence from openly available data; therefore, this study does not require ethical approval. This scoping review will be the first to collect evidence that has investigated the underlying drivers of obesity in Africa. By identifying gaps in the current body of literature, this study can influence forthcoming research on obesity in Africa. The results that will be generated from this review will be shared with the public through peer-reviewed publications and national and international conferences.

**Acknowledgements** We would like to acknowledge the South African Medical Research Council for proving the office facilities to undertake this project.

**Contributors** AJ wrote the first draft of the protocol. CI, SD, AWO and CSW critically revised the intellectual content of the manuscript.

**Funding** World Health Organization Region for Africa (grant number: N/A).

**Competing interests** None declared.

**Patient and public involvement** Patients and/or the public were not involved in the design, or conduct, or reporting or dissemination plans of this research.

**Patient consent for publication** Not required.

**Provenance and peer review** Not commissioned; externally peer reviewed.

**Data availability statement** The data that will be generated from this paper will be made publicly available.

**ORCID iDs**
Anelisa Jaca http://orcid.org/0000-0002-9814-8374
Chinwe Iwu http://orcid.org/0000-0003-0765-7497
Charles Shey Wiysonge http://orcid.org/0000-0002-1273-4779

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
