## [Reviewer comments · BMJ Open]

ARTICLE DETAILS

TITLE (PROVISIONAL)	Understanding the underlying drivers of obesity in Africa: A scoping review protocol.
AUTHORS	Jaca, Anelisa; Iwu, Chinwe; Durão, Solange; Onyango, Adelheid W.; Wiysonge, Charles

VERSION 1 – REVIEW

REVIEWER	James Bentham University of Kent, UK
REVIEW RETURNED	09-Jul-2020

GENERAL COMMENTS	This will be a valuable study on an important topic, and I think the study design will be effective. However, several possible drivers of obesity are not mentioned, e.g., smoking status, alcohol consumption, and urbanisation, and so these should be included in the study. My other comment is that statistical methods are not discussed. Do the authors intend to carry out any quantitative synthesis of the results (e.g., meta-analyses), or will the results from previous studies simply be summarised? This should be discussed explicitly in the text.
--

REVIEWER	Rosemary Geddes University of Edinburgh, Scotland, UK
REVIEW RETURNED	14-Jul-2020

GENERAL COMMENTS	General comments: This is an important topic given globalisation and the increasing epidemic of NCDs in Africa. This has been highlighted by the current COVID-19 pandemic, given that NCDs such as hypertension, cardiovascular disease and obesity have been risk factors for poor outcomes. The authors propose using the Arksey & O'Malley scoping review methodological outline which is an appropriate choice. It would be good to state the time period and dates that the scoping study will be undertaken, given that they are usually rapid reviews. Specific comments: Pg 5, Line 15/16 – 'middle-aged adults' traditionally are aged 36-55 years, and would not include 15-35 years, as is implied in this statement. Decide if writing in UK or US English. 'Meters' (Pg5, Line 36/37) and 'behaviors' (pg 6, line 7) is US spelling. Page 6, 2nd paragraph and pg 8 – You have defined obesity and overweight, but it is just as important to define 'physical inactivity'
---

	and 'sedentary behaviour'. I am aware of definitions for UK and Australia. Pg 8, 9, 10 – Inclusion criteria and 'exposures' – I believe that in order to capture papers on the full drivers, the search should be very wide at first. By only searching for two key areas (obesity terms and then African context names), you may miss studies which focus predominantly on physical inactivity/activity and sedentary behaviour. It might be worth doing a third key search for terms similar to these, and then combining that search with Boolean operator 'AND' with your #2. Page 11, Data charting and collating – You mention thematic areas, namely socio-economic, behavioural and medication. This seems fairly simplistic, given the complexity of drivers of obesity. It would be important to consider thematic areas in a bit more detail by organising into social & individual psychology, physical environment, food production & consumption etc. This would prompt exploring important driver areas such as culture, beliefs, habits, the built environment and access to green space. I would recommend that you take a look at the famous Foresight Report figure 5.2 (The full obesity system map with thematic clusters) for guidance or ideas. See https://assets.publishing.service.gov.uk/government/uploads/system/uploads/attachment_data/file/287937/07-1184x-tackling-obesities-future-choices-report.pdf Overall this is a good scoping study protocol.
--	---

VERSION 1 – AUTHOR RESPONSE

Reviewers' comments	Response
This will be a valuable study on an important topic, and I think the study design will be effective. However, several possible drivers of obesity are not mentioned, e.g., smoking status, alcohol consumption, and urbanization, and so these should be included in the study.	Thank you for your comment, we have added other factors contributing to obesity "In addition to dietary and sedentary patterns, other contributing factors to the high prevalence of obesity include smoking, alcohol consumption, urbanization and commonly used medications". Pages 4 and 6, line 45 and 102.
My other comment is that statistical methods are not discussed. Do the authors intend to carry out any quantitative synthesis of the results (e.g., meta-analyses), or will the results from previous studies simply be summarised? This should be discussed explicitly in the text.	Thank you, we have added, "We will therefore conduct a narrative synthesis of the underlying drivers of obesity in the African setting". Page 5, line 61.
The authors propose using the Arksey & O'Malley scoping review methodological outline which is an appropriate choice. It would be good to state the time period and dates that the scoping study will be undertaken, given that they are usually rapid reviews.	Thank you for your constructive comments, we have now stated, "We will thus conduct a scoping review, from the 1st of September 2020 to the 26th of February 2021, in order to map the crucial concepts and bring together the existing evidence on the underlying drivers of obesity in the African context". Page 5, line 75.
Pg 5, Line15/16 – 'middle-aged adults' traditionally are aged 36-55 years, and would	Thank you, we have now revised the sentence as This review will include obese and overweight

not include 15-35 years, as is implied in this statement.	individuals of all ages, i.e., children (between ages 2 to 12 years), adolescents (13 to 17 years), young adults(18 to 35 years), older adults (36 to 55 years) and the elderly (56 years and older)". Pages 6 and 7, lines 96 and 107.
Decide if writing in UK or US English. 'Meters' (Pg5, Line 36/37) and 'behaviors' (pg 6, line 7) is US spelling.	Thank you, we have changed the default language to the US English.
Page 6, 2nd paragraph and pg 8 – You have defined obesity and overweight, but it is just as important to define 'physical inactivity' and 'sedentary behaviour'. I am aware of definitions for UK and Australia.	Thank you for the comment, we have now defined physical activity and physical inactivity or sedentary behavior as "Physical activity is defined as any bodily movement that demands energy use, e.g., through walking, cycling, sport activities and doing house chores. On the other hand, physical inactivity or sedentary behavior is a lifestyle identified by an energy expenditure less than 1.5 metabolic equivalents (METs) while in a sitting, reclining or lying position". Page 4, line 40.
Pg 8, 9, 10 – Inclusion criteria and 'exposures' – I believe that in order to capture papers on the full drivers, the search should be very wide at first. By only searching for two key areas (obesity terms and then African context names), you may miss studies which focus predominantly on physical inactivity/activity and sedentary behaviour. It might be worth doing a third key search for terms similar to these, and then combining that search with Boolean operator 'AND' with your #2.	Thank you for the suggestion, we have taken note of this and will design a comprehensive search when conducting the study. We have added the search terms: physical inactivity OR physically inactive OR sedentar*. Pages 7 and 8, lines 119 and 132.
Page 11, Data charting and collating – You mention thematic areas, namely socio-economic, behavioural and medication. This seems fairly simplistic, given the complexity of drivers of obesity. It would be important to consider thematic areas in a bit more detail by organising into social & individual psychology, physical environment, food production & consumption etc. This would prompt exploring important driver areas such as culture, beliefs, habits, the built environment and access to green space. I would recommend that you take a look at the famous Foresight Report figure 5.2 (The full obesity system map with thematic clusters) for guidance or ideas.	Thank you for the suggestion, we will explore the factors that contribute to obesity, in depth. We have revised the sentence to "These will include demographic (age and gender), environmental (suburbs and rural areas), food availability (high calories and energy dense food), socio-economic (education level, employment status and income level), behavioural (e.g., physical inactivity, diet and alcohol consumption) and medication related drivers". Page 9, line 158

VERSION 2 – REVIEW

REVIEWER	James Bentham University of Kent, UK
REVIEW RETURNED	19-Aug-2020

GENERAL COMMENTS	My only comment is that the references should be checked during the proof stage. Ref 8 in the introduction refers to a paper on Bangladesh rather than Africa as stated. Also, ref 27 is to the wrong paper - it should be to Arksey and O'Malley.
--

REVIEWER	Rosemary Geddes University of Edinburgh UK
REVIEW RETURNED	03-Sep-2020

GENERAL COMMENTS	This is an improved version of the original and ready for acceptance, I believe.
--